# Interactions of SARS-CoV-2 with Human Target Cells—A Metabolic View

**DOI:** 10.3390/ijms25189977

**Published:** 2024-09-16

**Authors:** Wolfgang Eisenreich, Julian Leberfing, Thomas Rudel, Jürgen Heesemann, Werner Goebel

**Affiliations:** 1Structural Membrane Biochemistry, Bavarian NMR Center (BNMRZ), Department of Bioscience, TUM School of Natural Sciences, Technical University of Munich, Lichtenbergstr. 4, 85747 Garching, Germany; julian.leberfing@tum.de; 2Chair of Microbiology, Biocenter, University of Würzburg, 97074 Würzburg, Germany; thomas.rudel@biozentrum.uni-wuerzburg.de; 3Max von Pettenkofer Institute, Ludwig Maximilian University of Munich, 80336 München, Germany; heesemann@mvp.lmu.de (J.H.); goebel@biozentrum.uni-wuerzburg.de (W.G.)

**Keywords:** SARS-CoV-2, COVID-19, long COVID, metabolic reprogramming, persistence, host-directed therapies

## Abstract

Viruses are obligate intracellular parasites, and they exploit the cellular pathways and resources of their respective host cells to survive and successfully multiply. The strategies of viruses concerning how to take advantage of the metabolic capabilities of host cells for their own replication can vary considerably. The most common metabolic alterations triggered by viruses affect the central carbon metabolism of infected host cells, in particular glycolysis, the pentose phosphate pathway, and the tricarboxylic acid cycle. The upregulation of these processes is aimed to increase the supply of nucleotides, amino acids, and lipids since these metabolic products are crucial for efficient viral proliferation. In detail, however, this manipulation may affect multiple sites and regulatory mechanisms of host-cell metabolism, depending not only on the specific viruses but also on the type of infected host cells. In this review, we report metabolic situations and reprogramming in different human host cells, tissues, and organs that are favorable for acute and persistent SARS-CoV-2 infection. This knowledge may be fundamental for the development of host-directed therapies.

## 1. Introduction

Severe acute respiratory syndrome coronavirus 2 (SARS-CoV-2) belongs to the large group of positive-sense, single-strand RNA (+ssRNA) viruses [1,2], which includes some of the most human-pathogenic viruses, like polio virus, hepatitis C virus, dengue virus, zika virus, rhinoviruses, noroviruses, and—closely resembling SARS-CoV-2—severe acute respiratory syndrome coronavirus (SARS-CoV) and Middle East respiratory syndrome (MERS) coronavirus [3,4,5,6]. The RNA genome sizes of these RNA viruses differ considerably. While the genome size of many +ssRNA viruses, including polio virus and hepatitis E virus, is only about 7.5 kb, that of the three ß-coronaviruses mentioned above (the largest among the +ssRNA viruses) is approximately 30 kb [7]. The successful establishment of viral infections in human cells requires several steps, including the provision of suitable receptors (and eventually co-receptors), the uptake of the virus and the release of the viral genome into the cytoplasm of the host cell, the blockage of antiviral responses, the replication of the viral genome, the expression of the genes encoding virus-specific products, the assembly of new virus particles, and release from the cell. For reviews, see [8,9]. A host-cell metabolism adapted to the needs of the respective virus is crucial for the successful implementation of these steps that are necessary for efficient virus production [10,11,12]. After the virus has entered a host cell, it can either encounter a cellular metabolism that is suitable for its efficient replication or it can reprogram an existing metabolism that is less favorable for its replication. In a previous review, we described the adaption of the host cell metabolism upon infection via several DNA and RNA viruses in comparison with the metabolic changes triggered through infection with obligate and facultative intracellular bacteria [13]. Referring to this previous review and the discussion therein, we now focus on the metabolic programming of different host cells, which allows SARS-CoV-2 replication and proliferation (or persistence). 

To facilitate the understanding of the metabolic changes that occur when SARS-CoV-2 infects different host cells, we provide, in Appendix A, an overview of the relevant metabolic pathways, and in Appendix A, we illustrate their regulation in human cells. Generally, cell metabolism consists of two branches, catabolism and anabolism, both of which are necessary for maintaining cell viability and growth. The regulation of these metabolic pathways is performed in human cells through a complex network of regulatory factors and pathways [14,15,16] (partly shown in Appendix A), which can interact and functionally influence each other under specific nutritional and growth conditions. Specific metabolic pathways and their regulation are essential for viral replication and proliferation. Since these viral processes require large amounts of amino acids, nucleotides, and fatty acids/lipids for the production of the viral macromolecules, i.e., RNA (and DNA in the case of DNA viruses), proteins, and lipid-containing membrane structures, the respective metabolic pathways have to be active already or have to be activated through the virus and geared to the needs of the virus. The way in which this task is achieved can vary greatly between viruses. In this review, we sought to summarize what is known about these processes in the case of SARS-CoV-2. 

## 2. SARS-CoV-2 Proteins Involved in the Infection Cycle and Their Interactions with the Host-Cell Metabolism

The genome of SARS-CoV-2 carries information for four structural proteins (SPs), the spike glycoprotein (S), the envelope protein (E), the membrane protein (M), and the nucleocapsid protein (N) for 16 non-structural proteins (NSP1–16) and for 9 small accessory proteins (ACPs) [17,18,19]. For details regarding the structure and function of these SARS-CoV-2 proteins, see [6,17,20,21,22,23]. In the following subsections, we consider mainly those viral proteins that may affect (i) the entry of SARS-CoV-2 into host cells, (ii) the metabolism of the infected host cells, and (iii) the exit of intact virus particles.

### 2.1. Entry of SARS-CoV-2 into Host Cells via the Interaction of S Protein with ACE2 and Other Receptors

Of the four SPs whose functions are largely known [19,24], the S protein has received the most attention, as it is primarily responsible for the entry of SARS-CoV-2 into host cells [25,26,27]. The transmembrane S protein forms trimers in the virus envelope. After its biosynthesis, the S protein is cleaved in the Golgi via furin into subunits S1 and S2. The two subunits remain in the viral envelope after the virus has left the cell. The S1 subunit carries the receptor-binding domain (RBD) for the receptor angiotensin-converting enzyme 2 (ACE2) that is expressed in cells of many organs [28]. To trigger the virus’s entry into the host cell by fusing the viral envelope with the host membrane, the host transmembrane serine protease 2 (TMPRSS2) cleaves off a short internal fragment (130 amino acids) of the S2 subunit, creating S2′, which triggers fusion, pore formation, and the entry of the encapsulated virus into the cell [27,29]. In addition to this preferred, rapid entry pathway, the glycosylated S protein is also able to interact with C-type lectin receptors and other receptors (see the next paragraph). The uptake of the virus is carried out through the energy-consuming endocytic pathway and the delayed activation of S′/membrane fusion, e.g., via cathepsin L, allowing the release of the encapsulated virus. An important requirement for triggering the virus’s entry via membrane fusion or endocytosis is a sufficient density of cholesterol in the host-cell membrane for the formation of lipid rafts, which are the functional platforms for endocytosis. The dependence of SARS-CoV-2 infection on cholesterol-rich lipid rafts has been repeatedly demonstrated [30,31,32,33,34].

In addition to the primary interaction with ACE2, the S protein—due to its heavy glycosylation with mannosylated N-glycan and O-glycan moieties [35,36,37]—can also interact with several other non-immune- and immune-cell surface receptors that recognize glycosylated proteins. These receptors include (among others) the glucose-regulated protein 78 (GRP78), the C-type lectin receptors (CLRs) DC-sign (also termed CD209) and L-sign (CD209L), and the mannose receptor (MR), as well as the toll-like receptors TLR1, TLR4, and TLR6 [38,39,40]. GRP78, DC-sign (CD209), and (with lower efficiency) the related L-sign (also termed CD209L) appear to allow the entry of SARS-CoV-2 even into human cells that lack ACE2, e.g., endothelial cells (EnCs) [41,42]. We mention the interaction of the S protein with these additional receptors because the activation of these receptors is linked to the nuclear factor “kappa-light-chain-enhancer” of activated B cells (NF-kB), which plays an important role in the regulation of genes involved in inflammation, immunity, and cell metabolism [43,44,45,46]. The metabolic alterations in SARS-CoV-2-infected human cells that are linked to NF-kB are discussed below.

### 2.2. Synthesis of Viral RNAs and Proteins

After the release of the (+) RNA genome of SARS-CoV-2 into the cytoplasm of the host cell, it is first translated into two polyproteins (ORF1a and ORF1b) that are co-translationally cleaved into the NSPs via proteases that are part of the polyprotein (e.g., NSP5). NSP12 represents the core RNA-dependent RNA-polymerase, while NSP7 and NSP8 serve as cofactors for the formation of the functional replication and transcription complex (RTC) [23,24]. The replication of the SARS-CoV-2 RNA genome—like that of the genomes of all +ssRNA viruses—involves the synthesis of a negative-strand RNA and the asymmetric RNA synthesis of positive-strand RNA over negative-strand RNA. This step also depends on multiple host factors [1,47]. Like for other coronaviruses, SARS-CoV-2 proteins form a network of double-membrane vesicles, where viral RNA synthesis occurs. This will affect the host cell’s fatty acid/lipid metabolism due to the increased demand for fatty acids during viral membrane synthesis. However, in particular, infections caused by +ssRNA viruses could also generally trigger lipid rearrangements in their hosts [48,49]. In these membrane vesicles, RTC copies and transcribes the viral (+) genome into (−) full-length and shorter (−) subset RNAs. These function as templates for the synthesis of the complementary (+) RNA genomes and (+) subset RNAs, which serve as mRNAs for the translation of the four SPs: S, M, E, and N. Some of these mRNAs are also used for translation into ACPs. The amount of newly synthesized (+) strand RNAs exceeds by far that of (−) strand RNAs, which are degraded during viral replication [18,22]. This biosynthetic performance, which is necessary for the efficient replication of the virus, requires large amounts of energy, ribonucleotides, and amino acids from the host cell. Thus, it places a great burden on the cellular translation machinery and the defense system. Indeed, shortly after the infection of a host cell with the virus, already, the fraction of virus-encoded protein synthesis to the total cellular protein synthesis increases sharply [50,51]. In the following, we primarily discuss the virus-encoded NSPs and ACPs that are involved in processes linked to metabolism and the defense of infected cells [21,23,24,52,53].

### 2.3. NSPs and ACPs Influencing Host-Cell Metabolismolism and Defense

The NSP1 protein generally affects the host-cell translation machinery by binding to the 40S ribosomal subunit. This interaction hinders the access of cellular mRNAs to the ribosome, but it still allows the access of the viral RNAs (for further details, see [24,52,54]). This also affects the number of enzymes involved in metabolism. The actual metabolic activity in SARS-CoV-2-infected cells depends more on the stability of these enzymes and less on the amounts of mRNAs for these enzymes. This may complicate the interpretation of metabolic activities based on transcriptome data, as it remains questionable how much measured cellular mRNAs are translated into functional enzymes (see below). NSP12, in combination with NSP7 and NSP8, is involved in the formation of the functional RTC and is, therefore, indispensable for the efficient replication and transcription of viral RNAs. The balance of cell metabolism is maintained through several master regulators, including hypoxia-inducible transcription factor 1α (HIF-1α) and p53, which influence the regulation of energy metabolism, oxidative stress, and amino acid metabolism by altering the balance between glycolysis and OXPHOS [55]. NSP5, the main viral protease, suppresses the transcriptional activity of p53 [56]. The overexpression of p53 reduces SARS-CoV-2 production; the repression of p53 via NSP5, therefore, induces SARS-CoV-2 production [56]. ORF3a apparently participates in the increased production of HIF-1α by inducing mitochondria-derived reactive oxygen species (ROS), which activate HIF-1α [24,57]. NF-kB is involved in the regulation of genes essential for glucose and lipid metabolism [46,58]. It is, therefore, likely that SARS-CoV-2 proteins activating NF-kB may also affect the host-cell metabolism of SARS-CoV-2-infected cells. NF-kB is targeted and activated through several SARS-CoV-2 proteins [56]. The strongest activation of NF-kB has been observed with ORF7a, which leads to the induced production of proinflammatory cytokines [59]. Like ORF7a, ORF7b, NSP6, and NSP13 can also interfere with NF-kB regulated pathways. In particular, the phosphorylation of STAT1/STAT2 and the nuclear translocation of these transcription factors are inhibited [24,60,61,62]. The phosphorylated STAT1/STAT2 proteins are decisive mediators of interferon Iα/ß (IFN-Iα/ß) signaling, and they trigger, in combination with IRF9, the expression of the IFN-stimulated genes (ISGs). The ISGs exhibit antiviral functions and are, therefore, essential components of the cellular antiviral response and adaptive immunity [63]. These viral proteins thus inhibit an important antiviral response of the host cell. 

### 2.4. Synthesis of Intact Virus Particles and Their Release from the Host Cell

Viral RNA replication takes place in virus-induced double-membrane vesicles [64]. This compartmentation protects viral RNA synthesis from the rest of the cellular environment. Virus assembly starts with the coating of (+) RNA genomes with N proteins. The thus-generated nucleocapsid structures bud into the endoplasmic reticulum/Golgi compartment, where they acquire a lipid bilayer that already contains the structural proteins S, M, and E. Finally, the virus particles are released from the infected cells by utilizing the lysosomal exocytosis pathway [24]. ORF3a seems to promote this process [65]. The assembly of the virus particles and their exocytosis-mediated release from the host cell is linked to the formation of additional lipid structures, which require efficient fatty acid/lipid biosynthesis via the host [18,24,66]. There are numerous indications that SARS-CoV-2 can affect these regulators, leading to the rewiring of metabolic programs and the dysregulation of other functions of the infected host cell [57,67,68,69]. Specific SARS-CoV-2 proteins seem to directly interact with these and other cell components involved in metabolism or in antiviral response mechanisms [19,24,70]. Here, we focused only on interactions that affect—directly or indirectly—cellular metabolism.

## 3. Metabolic Programs and Metabolic Reprogramming in Human Cells during In Vitro SARS-CoV-2 Infection

Obviously, the above-described processes necessary for the formation of intact SARS-CoV-2 particles require large amounts of nucleotides, amino acids, and fatty acids/lipids and, hence, a highly active metabolism of the infected host cell. Certain human cell lines, frequently used for in vitro studies with SARS-CoV-2 (see below), apparently meet the metabolic requirements for efficient viral replication already from the start. The cell lines most used to study the biochemical requirements for a SARS-CoV-2 infection in vitro include Vero E6, HEK-293, Calu-3, A549ACE2, Huh7, and Caco-2 cells. These cell lines either exhibit increased cell longevity without being immortal (Vero E6 and HEK-293 cells) or are immortal cancer cells (Calu-3, Huh7, A549ACE2, and Caco-2). Vero E6 cells express the ACE2 receptor but lack TMPRSS2. Instead, the entry of SARS-CoV-2 into Vero E6 cells seems to depend on endocytosis, followed by cathepsin-mediated S′ endosomal membrane fusion and the release of the encapsulated virus [71,72]. Infection can be significantly enhanced in Vero E6 cells transfected with human TMPRSS2 [73]. The Calu-3, Huh7, A549ACE2, and Caco-2 cell lines express ACE2 and TMPRSS2, and they can be efficiently infected by SARS-CoV-2 [72,74,75,76]. All these cell lines readily grow not only in standard culture medium with bovine calf serum (FCS) but also in serum-free medium supplemented with defined carbon sources, such as glucose and Gln. Their central metabolic pathways, including glycolysis, PPP, glutaminolysis, and the TCA cycle are highly active under these growth conditions [77,78]. The SARS-CoV-2 that infects these cells, therefore, encounters a fully active cell metabolism, which the virus can directly use or further adapt (reprogram) specifically for its replication. 

### 3.1. SARS-CoV-2 Infection of Vero E6 Cells

Vero E6 cells infected with SARS-CoV-2 showed an increased glucose uptake and increased levels of the glucose transporters GLUT1, GLUT3, and GLUT4 compared to non-infected cells [79]. The significantly increased amounts of key glycolytic enzymes including hexokinase 2 (HK-2), phosphofructokinase 1 (PFK-1), and pyruvate kinase isoenzyme 2 (PKM-2), as well as an enhanced level of lactate dehydrogenase (LDH), combined with an elevated level of secreted lactate, indicated that the metabolism of the virus-infected Vero cells is shifted towards aerobic glycolysis (see Figure 1 for a scheme). As described above, this metabolic shift provides the infected cells [faster than the tricarboxylic acid cycle (TCA cycle)-dependent oxidative phosphorylation (OXPHOS)] with a high level of ATP that is required for the necessary anabolic performance (synthesis of nucleotides, amino acids, and fatty acids/lipids). The importance of the increased glucose metabolism for viral replication has been demonstrated by the fact that its inhibition with the glucose antimetabolite, 2-deoxyglucose (2-DG) drastically reduced the production of virus particles [79]. Amino acids can be provided either via an increased uptake of essential amino acids, the de novo biosynthesis of non-essential amino acids, or the degradation of cellular proteins. The supply of essential amino acids via the degradation of cellular proteins is less likely. Accordingly, the SARS-CoV-2 infection of Vero cells exhibited limited autophagy and the activation of autophagy inhibited SARS-CoV-2 propagation [80]. The viral ORF3a, ORF7a, and NSP15 seem to be involved in the autophagy inhibition of virus-infected Vero cells [81,82]. In addition, the NSP1-dependent host-protein translational shutoff [83] might block the synthesis of autophagy-initiating proteins. This limited autophagy possibly ensures that the destruction of virus particles is prevented. A high ATP/AMP ratio and high levels of a mammalian target of rapamycin complex 1 (mTORC1) activity, maintained throughout the SARS-CoV-2 infection of Vero cells, guarantee an efficient uptake of amino acids. Metabolomics studies [80], indeed, showed significantly higher levels of all amino acids in SARS-CoV-2-infected Vero cells compared to mock-treated control cells. Elevated levels of all nucleoside triphosphates (NTPs) required for the synthesis of the genomic and mRNAs of SARS-CoV-2 were also observed in virus-infected Vero cells, suggesting enhanced nucleotide synthesis. Furthermore, the elevated levels of 1,3-bisphosphoglycerate and sedoheptulose-7-phosphate in SARS-CoV-2-infected Vero cells were indicators of induced glycolysis and the pentose phosphate pathway (PPP), respectively. Aerobic glycolysis, as induced via SARS-CoV-2 infection in Vero cells, normally leads to reduced activity of the mitochondrial acetyl-CoA-driven TCA cycle and subsequently to a reduction in NADH/H^+^-dependent ATP production via OXPHOS (which is compensated for via aerobic glycolysis). However, the TCA cycle delivers important intermediates for the synthesis of nucleotides, lipids, and amino acids that are indispensable to producing virus particles. Through metabolic profiling, using differently ^13^C-labeled glucose isotopomers, Mullen and colleagues [69] could indeed show that glucose enters the TCA cycle by producing oxaloacetate (OAA) via increased pyruvate carboxylase (PC) expression. This ATP-dependent anaplerotic reaction, therefore, seems to play an important role in the development of SARS-CoV-2. Using [U-^13^C_5_]-Gln, the same authors further showed that SARS-CoV-2-infected Vero cells use glutaminolysis for the production of α-ketoglutarate (α-KG) but maintain either the reductive carboxylation of α-KG, leading to isocitrate (Icit), or the oxidative TCA cycle reaction, leading to succinyl-CoA (Suc-CoA) in subsequent reactions (see Figure 1). They also reported that the infection increases mTORC1 activity, as required for SARS-CoV-2 replication in these host cells. Moreover, at an early time of infection [(8 h post-infection (pi)], SARS-CoV-2 hijacks the host’s folate and one-carbon metabolism (C1 in Figure 1) and thus supports de novo purine synthesis [84]. This suggests that de novo purine synthesis is particularly important for extensive viral RNA production at the beginning of a SARS-CoV-2 infection. Not surprisingly, the inhibition of folate metabolism (via methotrexate) and C1 strongly reduced SARS-CoV-2 replication in these host cells [85,86].

### 3.2. SARS-CoV-2 Infection of Calu-3 Cells

Calu-3 cells are frequently used as a model for the airway epithelium in different SARS-CoV-2 studies, including those dealing with the re-programming of cell metabolism through SARS-CoV-2 [87]. Calu-3 cells grow—albeit more slowly than Vero 6 cells—in serum-containing culture media and in defined serum-free MEM media supplemented with glucose and/or glutamine (Gln). In contrast to Vero cells, Calu-3 cells show unlimited growth as cancer cells [88]. The central carbon metabolism is—as expected for cancer cells—mainly characterized by glucose and Gln consumption, and the major pathways fueled via these nutrients are aerobic glycolysis, the PPP, input into the TCA cycle through Pyr carboxylation, and the reductive conversion of α-KG (derived from Gln via glutaminolysis) to Icit (see also Figure 1) [88]. High intracellular glucose levels, supported by the GLUT1 transporter-induced ACE2 expression in Calu-3 cells and the inhibition of GLUT1, decreased ACE2 expression [89]. This could be an important finding, as it possibly explains why high cellular glucose levels, as evident in diabetes patients, lead to increased SARS-CoV-2 infections (for a recent review, see [90]). However, increased rates of SARS-CoV-2 infections in diabetes patients may also result from the enhanced uptake of the virus due to low serum levels of 1,5-anhydro-D-glucitol [91]. SARS-CoV-2 replication in Calu-3 cells depends, similar as in Vero cells, on aerobic glycolysis and glutaminolysis; the inhibition of these pathways limited viral production [92]. Proteomics analysis also showed that most of the significantly up-regulated proteins in the infected cells belonged to glycolysis, fructose and mannose metabolism, the PPP, amino acid biosynthesis, and nucleotide biosynthesis. Most of the significantly downregulated proteins belonged to the TCA cycle, OXPHOS, and FAO. Among the down-regulated proteins were all the mitochondrial enzymes of the TCA cycle, whereas cytosolic enzymes that produce or convert TCA cycle intermediates outside the mitochondria were up-regulated in the infected Calu-3 cells, including ACL, isocitrate dehydrogenase 1 (IDH1), cytoplasmic aconitase 1 (ACO1), and malate dehydrogenase 1 (MDH1) [92]. This suggested that intermediates of the TCA cycle, required especially for the biosynthesis of certain amino acids and fatty acids/lipids, are produced through anaplerotic reactions, i.e., Pyr carboxylation catalyzed via PC to OAA and the reductive carboxylation of α-KG [derived from glutamate (Glu)], leading to Icit/Cit and, subsequently, to acetyl-CoA (via ACL). The significant up- and down-regulation of the metabolic pathways, especially observed in Calu-3 cells, were in accord with the observed modulations of the protein kinase B/mammalian target of rapamycin/HIF-1α (Akt/mTOR/HIF-1α) signaling pathways triggered via SARS-CoV-2 infection in Calu-3 and other human cells, as described in previous studies [93,94].

### 3.3. SARS-CoV-2 Infection of A549 Cells

A549 cells, like Calu-3 cells, represent a human non-small-cell lung cancer (NSCLC) cell line that is commonly used for the study of respiratory infections. A549 cells express negligible levels of ACE2 and no TMPRSS2. Thus, A549 cells are per se poorly permissive to infection with SARS-CoV-2 [71]. To allow the better entry of SARS-CoV-2 into A549 cells, this cell line has been stably transfected with the human ACE2 gene, yielding A549ACE2 [95]. To gain a comprehensive view of the metabolic programming of host cells with SARS-CoV-2 infection, extensive comparative transcriptomic analysis (obtained via RNAseq) with SARS-CoV-2-infected A549ACE2 and Calu-3 cell lines was performed [96,97]. A highly significant downregulation of differentially expressed genes (DEGs), involved in the TCA cycle, in OXPHOS and FAO, was observed in both cell lines, which is in line with the above-described proteome data for SARS-CoV-2-infected Calu-3 cells. The observed significant downregulation of the DEGs involved in glycolysis, fructose, and mannose metabolism and in PPP (especially in A549ACE2, and also in Calu-3) was, however, unexpected and in contrast to the activation of these metabolic pathways reported for SARS-CoV-2-infected Vero cells (see above) observed through proteome/metabolome analysis. Also, in contradiction with the metabolome data was the downregulation of the DEGs involved in pyrimidine and purine metabolism, as well as in fatty acid synthesis (FAS). These metabolites are required for viral replication, and they have been shown to be increased through metabolome data. These discrepancies may indicate that the level of transcripts of the genes involved in catabolic and anabolic pathways does not necessarily correlate with the dynamics and the level of the intermediates and end products of such pathways since the conversion of the primary transcripts to the active enzymes proceeds not necessarily in a colinear manner. It may involve several posttranscriptional and post-translational steps, including the stability of the primary mRNA, its possible posttranscriptional modification(s), its translation efficiency, and, finally, the possible regulation of its enzymatic activity via a variety of factors.

### 3.4. SARS-CoV-2 Infection of Caco-2 Cells

Caco-2 cells represent a colon carcinoma cell line that expresses high levels of ACE2 and TMPRSS2. Since SARS-CoV-2 infections in humans may also affect the gastrointestinal (GI) tract [98], this colon epithelial cell line could be a relevant model system for SARS-CoV-2 GI infections. Caco-2 cells are as highly permissive to SARS-CoV-2 as the NSCLC line Calu-3 [87,99]. Even a short exposure with low viral doses leads to efficient infections of Caco-2 cells. The SARS-CoV-2 infection of Caco-2 cells activated the epidermal growth factor receptor/phosphoinositide-3-kinase/Akt/mammalian target of the rapamycin complex 1 (EGFR/PI3K/AKT/mTORC1) pathway [100,101], thereby increasing the synthesis of products required for viral replication and inhibiting autophagy in SARS-CoV-2 infected cells. In infected Caco-2 cells, the production of proteins involved in the TCA cycle and OXPHOS was reduced, whereas the synthesis of enzymes involved in glycolysis, as well as in the production of nucleotides, fatty acids, and cholesterol, was enhanced. Proteomics data also showed an increased expression of transketolase (TKT), an enzyme involved in the non-oxidative arm of the PPP leading to R5P. The TKT inhibitor benfooxythiamine (BOT) reduced SARS-CoV-2 infection in Caco-2 cells [102]. The inhibition of fatty acid metabolism strongly impaired SARS-CoV-2 replication by altering the membrane-containing replication center of the virus in the infected host cells [103]. 

### 3.5. SARS-CoV-2 Infection of Huh-7 Cells

Huh-7 cells represent a cell line derived from human hepatoma. Huh-7.5 is a subline of Huh-7 with a defect in innate antiviral signaling. Both cell lines have also been widely used as hepatocyte in vitro models for SARS-CoV-2 infection [104]. Huh-7 cells resemble liver tumor cells in their metabolism [105] with a central role of glucose and Gln. ACE2 is expressed at low levels in Huh-7 and Huh-7.5 [106,107]. Nevertheless, several studies demonstrated that Huh-7 and Huh-7.5 are susceptible to infection with SARS-CoV-2 [99,104,107,108]. The efficiency of virus production in Huh-7 cells is moderate but significantly enhanced when the cells are pre-infected with hepatitis C virus (HCV) [107]. The HCV infection increased the expression of ACE2, which may facilitate the entry of SARS-CoV-2 in the Huh-7(HCV) cells—as argued by the authors. However, one should also consider that HCV massively alters the metabolism of hepatocytes [109], and this altered metabolism could favorably affect the replication of SARS-CoV-2 in Huh-7(HCV) cells.

### 3.6. SARS-CoV-2 Infection of HUVEC Cells

HUVEC cells represent human umbilical-vein endothelial cells and are, therefore, used as an in vitro model for SARS-CoV-2 infections of EnCs. ACE2 and TMPRSS2 are expressed in EnCs [110,111,112], as well as other serine proteases [113], suggesting that the SARS-CoV-2 infection of EnCs is theoretically possible. However, the expression of ACE2 and TMPRSS2 proteins in HUVECs appeared to be much lower than in the epithelial Calu-3 cells used as a positive control [114]. These authors also found that HUVECs are not productively infected by SARS-CoV-2. However, senescent HUVECs were, in contrast to young cells, highly susceptible to SARS-CoV-2 infections [115]. The entry of the virus into the senescent HUVECs seems to occur in an ACE2-independent manner. The infection of the senescent HUVECs led—compared to young cells—to the expression of a large number of genes, including genes whose products seem to be involved in dysfunctions of EnCs in vivo that may contribute to the increased inflammation and thrombosis observed in severe COVID-19 cases. Based on the present data, the ACE2-mediated uptake of SARS-CoV-2 does not seem to be the main reason for the different susceptibility of SAR-CoV-2 infection in senescent and young HUVECs. The question, therefore, arises as to whether the metabolism of senescent HUVECs may favor the obviously more efficient viral replication in these cells. Senescence is clearly associated with changes in cell metabolism triggered by a complex response to a variety of environmental stresses [116]. Enhanced glycolysis appears to be a hallmark of senescent cells, and it has been suggested that senescent cells may favor aerobic glycolysis to provide energy and precursors to the high demand for protein, lipid, and other cellular components [116,117]. By comparing the metabolism of replicative senescent HUVECs (passage 25) with young HUVECs (passage 1), Stabenow and colleagues [118] could indeed show that the senescent cells exhibited higher glycolytic activity and lactate production associated with an enhanced expression of LDH-A. Yet, PDH is also activated, and PDHK is downregulated in senescent cells, indicating acetyl-CoA production, an active TCA cycle, and, hence, mitochondrial respiration that may promote the production of ROS. The significantly enhanced glucose uptake and the increased glycolytic activity of senescent HUVECs compared to young HUVECs may, thus, support the replication of SARS-CoV-2 in senescent HUVECs. In quiescent EnCs, FAO leads to the regeneration of NADPH, which appears to be important for maintaining the redox balance [119]. This type of cell metabolism is also unfavorable for SARS-CoV-2 replication and could contribute to the inability of this virus to infect quiescent EnCs [114]. 

### 3.7. SARS-CoV-2 Infection of Monocytes and Macrophages

Monocytes and macrophages (MPs) are among the first immune cells to interact with SARS-CoV-2. These phagocytes are able to take up the virus and thereby contribute to its spread [120,121]. Monocytes circulating in the blood can differentiate into subtypes, of which the classic activated M1 and the alternatively activated M2 MPs are best characterized. However, this known binary M1/M2 classification may not be sufficient to capture the diversity of MPs in vivo [122]. The metabolism of M1 MPs depends on glucose as a major energy source, the induction of GLUT1, increased glycolysis, and the enhancement of lactate production. This type of metabolism is also named “Warburg-like” metabolism, well known in cancer cells to describe a metabolic shift where cells rely more on glycolysis than on OXPHOS for fast ATP generation, even in the presence of oxygen. PPP is also promoted under these conditions which supports the generation of R5P and NADPH, necessary for nucleotide biosynthesis. OXPHOS and FAO are downregulated. Glycolysis, PPP, and, to a lesser extent, the TCA cycle, yield the intermediates for the biosynthesis of specific amino acids, fatty acids/lipids, and nucleotides necessary for cell maintenance [123,124]. The specific hallmark of M1 MPs is their elevated production and secretion of the inflammatory cytokines/interleukins IL-1ß, IL-6, IL-23, and TNF-α [125,126]. The metabolism of M2 MPs depends mainly on FAO and OXPHOS. Glucose uptake and glycolysis are not mandatory if OXPHOS is functional, but it may become necessary if OXPHOS is compromised [127,128]. The M2 MPs exhibit anti-inflammatory activity by releasing IL-4, IL-10, and TGF-ß cytokines [126,129], and they play a decisive role in tissue repair [130,131].

Most studies on monocytes in combination with SARS-CoV-2 infection have been performed in vitro with the monocytic cell line THP-1 or with blood monocytes, which are part of the ”peripheral blood mononuclear cells“ (PBMCs) population obtained from SARS-CoV-2-infected patients and from healthy control persons [132,133]. The THP-1 cell line can be differentiated (like blood monocytes) into MPs via treatment with phorbol-12-myristate-13-acetate (PMA). The resulting resting (M0-like) MPs can be further differentiated in the direction of M1 and M2 phenotypes through incubation with lipopolysaccharide, LPS/IFN-γ, and IL-4/IL-13, respectively [134,135]. The infection of these M1- and M2-like THP-1 cells with SARS-CoV-2 showed that M1-like but not M2-like cells are able to efficiently take up SARS-CoV-2 and even allow the substantial generation of viral particles [134]. The authors suggested that the lower pH of the endo-lysosomal system of M1 MPs compared to that of M2 MPs might favor a more efficient release of the viral genomic RNA into the cytosol and its subsequent replication in M1 MPs. However, one should also consider the possibility that the different metabolic phenotypes of these cells (see above) could be responsible for the different capabilities of SARS-CoV-2 replication in these two THP-1 subtypes. 

Activated MPs from mouse bone marrow (BMDM) and peritoneal MPs, which can roughly differentiate into the subtypes M1 and M2, are targets of SARS-CoV-2 [136,137]. They exhibit the characteristic metabolic features described above [131,138]. The activated M1 subtype of these MPs is the major source of the excessive production of inflammatory cytokines, such as IL-6, IL-1ß, and TNF-α, that trigger tissue damage after SARS-CoV-2 infections. 

Pulmonary MPs are early targets encountered by SARS-CoV-2 [122,139,140]. This lung MP population consists mainly of two types of MPs: the tissue-resident alveolar MPs (AMs), which are the most abundant immune cells in the alveolus, and the interstitial MPs (IMs) which reside in the interstitium between the alveolar epithelium and capillary beds. Using an ex vivo lung-slice model derived from healthy humans, Wu and colleagues [141] could indeed show that activated IMs are prominent targets for SARS-CoV-2, while AMs were hardly infected by the virus. In the infected IMs, more than 60% of the cellular transcriptome consists of viral transcripts. The entry mechanism of SARS-CoV-2 into these MPs remains unclear, but it does not seem to require phagocytosis or ACE2-mediated uptake. The infected IMs, but not the AMs, induced an inflammatory cytokine/chemokine program. These results showed again that, in MPs, SARS-CoV-2 replication and the generation of mature viral particles are favored by the metabolism of the M1-like subtype.

### 3.8. SARS-CoV-2 Infection of Dendritic Cells

Dendritic cells (DCs) are the most potent antigen-presenting cells, and in this capacity, they bridge innate and adaptive immunity. DCs represent a heterogeneous family of immune cells that are derived from myeloid progenitors in the bone marrow, but they can also develop from blood monocytes (moDCs) under inflammatory stimuli [142]. Both types of DCs were used in studies analyzing the interaction of these immune cells with SARS-CoV-2. For recent reviews, see [143,144]. The reports about the entry and replication of SARS-CoV-2 in DCs are partly ambiguous [38,143,145,146]. DCs appear to express moderate levels of ACE2, and, therefore, the S protein-mediated entry of SARS-CoV-2 in DCs may be possible. In addition, S protein apparently recognizes still other receptors on the surface of DCs, and viral uptake could also occur through endocytosis [38]. The extent of replication of SARS-CoV-2 within DCs may depend on the metabolic state of the respective DC, which, in turn, is determined by the conditions of the surroundings of the DC within the human body.

### 3.9. SARS-CoV-2 Infection of Lymphocytes

B and T lymphocytes are the key players in adaptive immunity, and they play an important role in the development of protective immunity against SARS-CoV-2 infections [147,148]. Here, we focus on the metabolism of T and B cells and the interaction of these cells with SARS-CoV-2, as these lymphocytes apparently can be directly infected by the virus [149,150,151]. As shown by Shen et al. [150], activated T cells, mainly CD4^+^ T cells, were directly infected by SARS-CoV-2 in an ACE2-independent manner. Circumstantial evidence was provided that suggested that the lymphocyte function-associated antigen (LFA-1) could act as a receptor for SARS-CoV-2 to enter these cells. Several other receptor candidates have also been discussed [151]. Pontelli et al. [152] reported SARS-CoV-2 infections in CD4^+^ and CD8^+^ lymphocytes, in B-lymphocytes, and in monocytes as the most susceptible target cells. Notably, immune cells are in different metabolic states, depending on their differentiation and activation stage. A “Warburg-like” metabolism, as described above for M1-type macrophages, seems to be the optimal metabolic condition for the successful replication of SARS-CoV-2 after entry into such cells. It should be emphasized that the CD4^+^ T cells infected with SARS-CoV-2 provide these metabolic conditions best among the immune cells present in the PBMCs mixture.

## 4. Metabolic Programs and Metabolic Reprogramming in Human Tissues and Organs during SARS-CoV-2 Infection

From the previous discussion of the metabolic requirements for efficient replication of SARS-CoV-2 in different cells, the following picture emerges: enhanced glucose uptake followed by (aerobic) glycolysis, active PPP, reduced TCA cycle, and OXPHOS and the activation of anaplerotic pathways, based on the consumption of Gln (glutaminolysis) and fatty acids (FAO) yielding intermediates such as α-KG, OAA, Cit, and acetyl-CoA (see also Figure 1). This type of cellular metabolism apparently provides the necessary energy and metabolic intermediates required for efficient viral replication and blocks the antiviral responses of the host cells, such as the synthesis of type I IFNs, the subsequent induction of the antiviral ISGs, and the production of natural killer (NK) cells, which can inhibit viral infection and eliminate virus-infected cells. The situation is more complex for in vivo SARS-CoV-2 infections in which the virus meets tissues and organs consisting of various cell types performing different functions that require different metabolic activities. The crucial question is: How does a possible (ACE2^+^) target cell in a tissue attain this metabolic state suitable for the replication and propagation of SARS-CoV-2? There are several conceivable options:(i)The target cell carries out the appropriate metabolism right from the start, which seems to be the case for most of the described cell lines and some activated immune cells (and for cancer cells).(ii)In an in vivo cell population with largely metabolically resting cells, there are some (stochastically occurring) cells that carry out an appropriate metabolism that meets the requirements for viral replication. The occurrence of metabolic heterogeneity within the cell population in vivo is well established [153]. The initial replication of the virus in these few target cells may trigger a cascade of events that activate the metabolism of neighboring resting cells.(iii)The interaction of a viral surface protein with a suitable cell receptor can trigger signaling pathways, leading to metabolic changes that favor viral replication. The well-characterized interaction of the S protein of SARS-CoV-2 with the ACE2 receptor, which triggers the entry of the virus into the cell, does not seem to significantly affect the cellular metabolism. However, the S protein (and possibly other SARS-CoV-2 proteins) can also interact with additional receptors, such as GRP78, MR, DC-sign, and TLRs [42,154,155]. These interactions can activate signaling pathways and transcription factors, such as NF-kB and HIF-1α, which may contribute to the metabolic reprogramming of target cells [70,156,157,158]. Activated NF-kB, together with HIF-1α, not only regulates the expression of genes involved in immune response, inflammation, apoptosis, and cell survival but can also affect metabolic pathways, including glucose and lipid metabolism [46,58].

It is likely that, in vivo, all three options may differently contribute to the metabolic basics favorable for SARS-CoV-2 replication in possible (ACE2^+^) target cells. The extent of each option probably depends on the actual situation and physiological condition of the infected individual, and it could significantly determine the severity of an infection. 

### 4.1. The Nasal Epithelium and the Lower Conducting Airway as Target of SARS-CoV-2

The first tissue met by SARS-CoV-2 is the nasal epithelium (NE). The NE is composed of various cell types, including ciliated cells, goblet cells (GCs), and basal cells [159]. The expression of ACE2 is highest in GCs, followed by the ciliated cells and the basal cells. However, the cells that are primarily infected by the virus are ciliated cells, followed by GCs and eventually basal cells [160,161,162,163]. This already indicates that the marked tropism of SARS-CoV-2 for ciliated cells cannot be solely based on the ACE2 concentration on the cell surface. An interesting recent report suggested that mobile cilia and microvillar reprogramming are the decisive factors for the preference of ciliated epithelial cells (EpCs) for SARS-CoV-2 infection in NE [164]. But there are also obvious differences in the metabolism of the different cell types of the NE, which might play a role in this cell tropism. Ciliated EpCs are specialized differentiated cells in the respiratory epithelium that play a critical role in removing foreign particles from the airways, including viruses. For the formation and mobility of the cilia, these cells must maintain a highly active energy, protein, and lipid metabolism. This is achieved through a steady supply of nutrients, such as glucose, amino acids, and fatty acids, and induced catabolic (especially glycolysis) and anabolic (FAS and nucleotide biosynthesis) pathways [165]. Upon S protein/ACE2-mediated uptake into ciliated cells, SARS-CoV-2 may, therefore, hijack the active cellular metabolism and re-direct it for the needs of viral replication and propagation. Indeed, the SARS-CoV-2 infection induced a severe loss of cilia while actively replicating the virus [166,167,168]. GCs produce and secrete mucus, which serves as a lubricant to keep tissues from drying out and is responsible for trapping and removing foreign particles (including viruses) from the nasal cavity and other parts of the airway. The major activity of GCs is focused on the production of mucus that consists of an aqueous mixture of mainly glycoproteins (mucins) [159,169]. Using primary human airway epithelial (HAE) cells containing differentiated ciliated cells and GCs as a cell culture model, Pinto et al. [170] showed that ciliated cells are much more efficient than GC in replicating SARS-CoV-2. This clear tropism of SARS-CoV-2 for ciliated cells may be explained by the more suitable energy-oriented metabolism of the ciliated EpCs compared to the GC [171,172]. The basal cells are stem/progenitor cells, located in the basal layer of the NE, with the ability to self-renew and differentiate into ciliated cells and GCs, as well as other cell types [173,174]. Thus, they play an important role in the repair of the NE when ciliated cells and GCs are injured via SARS-CoV-2 infection. Basal cells seem to be more or less spared from infection by SARS-CoV-2, although the cells carry ACE2 and TMPRSS2 on their surface [175]. However, the data pertaining to this statement are inconsistent [160,168,176,177]. This is not surprising in view of the possible metabolic heterogeneity of this stem/progenitor cell population. Depending on the environmental conditions, part of this cell population might be in a strict quiescent state, while others are in a renewal or proliferation state [178]. In these different stages, they show different metabolic activities [179,180] that may, in part, be permissive of the replication of SARS-CoV-2.

### 4.2. The Lower Respiratory Tract as a Target of SARS-CoV-2

The lower respiratory tract (LRT) consists of the trachea, the bronchi, and, as major part, the lungs, including the alveoli and the associated blood vessels. The alveoli represent the terminal part of the bronchi, and they are the functional units of the lungs responsible for the gas exchange between the lungs and the bloodstream. The LRT and, here especially, the lungs, consist of numerous cell types that contribute to lung epithelial homeostasis and the gas-exchange function [181,182]. However, here, we only consider those cell types of the LRT that reportedly can act as target cells for SARS-CoV-2, including bronchial and alveolar EpCs, the EnCs of the blood vessels, and alveolar MPs [168,183,184,185]. In the bronchi, the ciliated cells are—as in the NE—the primary cell type infected by SARS-CoV-2, while the progenitor basal cells are again spared from infection [176,177]. The reason for the different infection courses of these bronchial cells could be the same as discussed above for the same cell types of the NE. The alveolar epithelium consists of two, structurally and functionally different alveolar cell types: type 1 (AT1) and type 2 (AT2). AT1 cells are thin and flat cells, they make up most of the alveolar surface, and they are mainly responsible for the gas exchange between the lungs and the bloodstream. AT1 cells are terminally differentiated cells derived from AT2 cells. AT2 are smaller and more cuboidal cells. They represent the primary progenitor cells for the alveolar epithelium and can differentiate into AT1 cells if necessary, e.g., upon an injury of the AT1 layer. The specific performance of the AT2 cells is the production and secretion of large amounts of surfactant, a mix of surfactant proteins, and phospholipids. Surfactant is required for reducing the surface tension in the alveoli, thus preventing their collapse during exhalation [186,187]. The metabolism of AT2 cells is geared towards supporting their specific functions (the production and secretion of surfactant, self-renewal, and differentiation) and designed to produce the necessary large amounts of energy, lipids, nucleotides, and proteins. It primarily relies on glucose and glycolysis for ATP production, but the TCA cycle and OXPHOS seem to also be active, and it relies on the PPP for the generation of NADPH and R5P [188,189]. FAO (especially of palmitate) can act as an alternative energy source in the case of glucose starvation [190]. The efficient production of ATP and NADPH is critical in the biosynthesis of the surfactant. Depending on the actual state of the AT2 cells (surfactant production, proliferation, and differentiation), the expression of the involved pathways may vary on the transcriptional and the enzymatic levels [189]; e.g., hyper-glycolysis in AT2 cells may support surfactant synthesis, but it may impair the differentiation of AT2 cells into AT1 cells [191]. Compared to the AT2 cells, the AT1 cells perform a low-level metabolism with little anabolic activity. The upregulation of metabolic activity, e.g., increased glycolysis, promotes cell aging or de-differentiation back to the AT2 phenotype [191]. The main cell type in the lungs that can become infected by SARS-CoV-2 is AT2 cells, whereas AT1 cells essentially remain uninfected [192,193]. However, single-cell RNA profiling showed that, even in the AT2 cell population, only a relatively small proportion of cells contain detectable amounts of mRNA encoding ACE2 [192,194]. But an even smaller fraction of AT2 cells was efficiently infected by SARS-CoV-2, as determined via SARS-CoV-2 specific antigens and mRNAs within AT2 cells [183,195]. This finding suggested, again, that it is not only the presence of the ACE2 receptor but, in addition, a proper metabolic basis of the target cell suitable for the replication of SARS-CoV-2. Since SARS-CoV-2 will hijack the metabolism of the infected cells by blocking the translation of the cellular mRNAs, but not of its own mRNAs (see above), it is not unexpected that the infection would lead to a reduction in the production of surfactant via AT2 cells with the subsequent pathological consequences [196,197]. In addition, AT2 cells are the main progenitor cells for the alveolar epithelium, similar to the basal cells for the NE. While the basal cells are, however, spared from SARS-CoV-2 infection and, hence, can restore the NE when damaged through the SARS-CoV-2 infection of ciliated and GCs, the partial destruction of the AT2 cells due to the SARS-CoV-2 infection can severely damage the alveolar epithelium due to failed restoration.

### 4.3. SARS-CoV-2 Infection of the Intestinal Epithelium

The intestinal epithelium (IE) is structurally organized into the crypt, including intestinal stem cells (ISCs), Paneth cells (PCs), and the villus, containing EpCs and GCs as major cell types. ISCs are multipotent and capable of both self-renewal and differentiation; they give rise to the other differentiated cells of the IE [198,199,200]. Organoids mimicking the IE can be generated in vitro from different pluripotent stem cells, which differentiate into the major cell types of the IE, namely EpCs, GCs, PCs, and enteroendocrine cells [201,202,203]. EpCs represent not only the most abundant cell type but also the metabolically most active cells in the IE. They absorb the breakdown products of digestion via specific transporters for carbohydrates (especially glucose), amino acids, and fatty acids [200,204]. Glucose and other carbohydrates are catabolized through glycolysis and fatty acids through FAO to generate energy. The TCA cycle and OXPHOS are also active, but glucose consumption and glycolysis seem to be the dominant pathways. Excess glucose is converted into glycogen and fatty acids into triglycerides, and they are stored in enterocytes, the most abundant form of the EpCs in the IE, or transported to other tissues [204,205]. PCs are highly specialized cells essential for the production and secretion of antimicrobial peptides (defensins) [206]. PCs and ISCs seem to be metabolically intermingled: PCs perform a more glycolytic metabolism, whereas ISCs rely more on an oxidative metabolism. Lactate generated via glycolysis in PCs is excreted and used by ISCs to support the oxidative metabolism. The differentiated GCs reside throughout the length of the IE and synthesize and secrete (like nasal GCs) high-molecular-weight glycoproteins (mucins). EpCs are also the most susceptible cell type of the IE for SARS-CoV-2 infections, as shown in vivo [207] and in various organoid models [208,209,210,211,212]. The infection is apparently promoted by the high levels of ACE2 and TMPRSS expressed by these cells [208,213]. However, it has been shown that the cellular levels of ACE2 are not proportional to the infectability of EpCs by SARS-CoV-2 and that even low levels of ACE2 are sufficient for the uptake of SARS-CoV-2 and efficient replication in EpCs [208,214]. Therefore, it is probably not only the ACE2 level but also the appropriate metabolism that make some EpCs a susceptible host cell type for SARS-CoV-2. In vivo, SARS-CoV-2 infection of the other IE cell types has not been convincingly shown [207,215]. ISCs, PCs, and GCs of the IE seem to express ACE2, albeit at a much lower level than EpCs [213,216]. Most studies analyzing the possible replication of SARS-CoV-2 in these IE cell types were performed in organoids containing mainly EpCs, GCs, PCs, and endocrine cells [208,211,212,217]. In all of these in vitro studies, the EpCs were efficiently infected with SARS-CoV-2. The infection of the other cell types is more controversial. The GCs of the IE remained uninfected [211,212], in contrast to the GCs of the respiratory epithelium [218]. The SARS-CoV-2 infection of endocrine cells was detected in one organoid study [211] but not in another one [212]. The infection of PCs was reported in both in vitro studies, while the in vivo infection of PCs and ISCs seems to be unlikely [219]. The different results of these infection studies could be due to differences in the metabolism of these cell types caused by the different culture conditions for organoids, which are certainly different from the in vivo conditions. This is obviously the case for the highly glycolytic PCs of the IE, which are metabolically linked to the oxidative ISCs in the crypt, and both are not infected by SARS-CoV-2 in vivo [219]. In organoids, the isolated PCs may carry out an unaffected glycolytic metabolism, which is better suited for the replication of SARS-CoV-2. This is a more hypothetical explanation since detailed studies on the metabolism of the various cell types in intestinal organoids are missing—at least to our knowledge. Generally, there is still only limited data concerning the metabolic adaptation of the intestinal tissue infected by SARS-CoV-2.

### 4.4. SARS-CoV-2 Infection of the Heart Tissue

The heart tissue is composed of different cell types, the most abundant of which are cardiomyocytes (CMs), fibroblasts (FBs), EnCs, and peri-vascular cells. CMs are muscle cells responsible for the contraction of the heart, FBs produce and maintain the extracellular matrix of the heart tissue, EnCs line the inner surface of the blood vessels in the heart, and pericytes, located around blood vessels, are involved in the regulation of blood flow. Increased cardiovascular symptoms are observed in a significant number of SARS-CoV-2-infected patients [220]. ACE2, the main receptor for SARS-CoV-2, is expressed in particular in CMs and, at a lower level, in FBs, EnCs, and pericytes [221], suggesting that these cell types could take up SARS-CoV-2. Indeed, pluripotent stem cell-derived CMs were efficiently infected by the virus [222]. The preferential infection of CMs was also observed in a similar heart-tissue model [223]. In addition, these authors could show that the metabolism of CMs changes from an oxidative state to a highly glycolytic state during infection, which apparently supports SARS-CoV-2 replication. The in vivo situation is less clear. Bailey et al. reported that SARS-CoV-2 directly infects CMs but not FBs and EnCs in patients with COVID-19 myocarditis [223]. In contrast, Bräuninger et al. [224] found—by analyzing 95 SARS-CoV-2 positive autopsy cases—that EnCs were frequently infected by SARS-CoV-2, whereas the infection of CMs was rarely detected, despite the presence of ACE2 and TMPRSS. Indeed, the metabolism of CMs and EnCs may vary from a more resting oxidative state to a more glycolytic state, depending on the actual conditions (differentiation, proliferation, and stress) [225,226,227,228]. CMs freshly obtained from stem cells (see above) are probably still in a proliferating glycolytic state and, hence, appropriate host cells for SARS-CoV-2. However, the nature of the metabolic response of heart tissue upon viral infection is basically unknown.

### 4.5. SARS-CoV-2 Infection of the Central Nervous System (Brain)

A relatively large proportion of COVID-19 patients suffer from neurological symptoms, suggesting that the central nervous system (CNS) can be invaded by SARS-CoV-2. The most likely access to the CNS is viral spreading through the olfactory mucosa, thus causing brain infections [229,230]. Indeed, autopsies of patients who died of COVID-19 showed sub-genomic SARS-CoV-2 RNA in the CNS, which is a sign of active viral replication in brain cells [231,232]. There are two main cell types present in the brain that are also the likely candidates for SARS-CoV-2 infections: neurons and non-neuronal glial cells (astrocytes, microglia, and oligodendrocytes). Astrocytes and microglia are the most abundant cells in the brain. Astrocytes are mainly involved in supporting neuronal functions, while microglial cells are the dominant resident immune cells acting as the immune system of the brain. Both neurons and glial cells carry out distinct but partially interacting metabolic processes that are essential for proper brain function [233]. 

Studies regarding the SARS-CoV-2 infection of brain cells have been performed with various neural models, including a post-mortem specimen of SARS-CoV-2 patients with neurological symptoms, human pluripotent stem cell-induced neural cell models, such as pure neurons, astrocytes, and microglia, as well as 3D organoids. The results of most of the earlier studies show that it is not clear or sometimes even contradictory [234,235] which cells of the brain are susceptible to SARS-CoV-2 [236]. The expression of ACE2 has been detected in cultured astrocytes and microglial cells but not in neurons, suggesting that astrocytes might be target cells for SARS-CoV-2 [237,238]. However, pluripotent stem cell-derived neurons also express ACE2 [230], and studies with 3D human brain organoids derived from pluripotent stem cells provided evidence that SARS-CoV-2 infects neurons but not astrocytes [239,240]. The data on the in vivo expression of ACE2 on glial cells and neurons are also still rather ambiguous [236,238]. In a recent study using in vivo brain samples from individuals with neurological symptoms (who died of SARS-CoV-2 infections) and in vitro cultured human neural stem cell-derived astrocytes infected with SARS-CoV-2, the authors showed that astrocytes are the main site of infection (and probably replication) for SARS-CoV-2 in the brain [241]. Interestingly, they also showed that the astrocytes do not express ACE2, but the efficient uptake of the virus occurs via the interaction of the spike protein with neuropilin-1 (NRP1), an alternative receptor for SARS-CoV-2 [242]. The preferential infection of astrocytes by SARS-CoV-2 was also reported by Andrews et al. in 2022 [243] using cortical organoids derived from human pluripotent stem cells (hPSCs), as well as from primary human cortical tissues. However, neurons and microglia did not seem to be directly infected by SARS-CoV-2. In contrast, human microglial clone 3 cells (HMC3) were infected by SARS-CoV-2, as shown by Jeong et al. in 2022 [244]. This virus-infected cell line showed an M1 phenotype that may support the replication of SARS-CoV-2, as described above for the replication of the virus in M1 macrophages. However, the authors also showed that, in K18-hACE2 transgenic mice, the microglia were also infected by the virus upon intranasal inoculation with SARS-CoV-2.

These partially ambiguous in vitro and in vivo results concerning the infection of brain cells by SARS-CoV-2 are not too surprising if one considers the complex metabolic interactions between brain cells and, in particular, between astrocytes and neurons that occur under in vivo conditions [233,245,246]. Astrocytes are mainly glycolytic cells that can even survive the inhibition of mitochondrial respiration by performing enhanced aerobic glycolysis [246]. Glucose is the preferred carbon and energy substrate, which is channeled into glycolysis and PPP, while the entry of Pyr into the TCA cycle is limited due to reduced PDH activity caused by phosphorylation through enhanced PDHK activity [247,248]. Excess Pyr is converted into lactate, which is secreted and which serves as a substrate for neurons (see below). The TCA cycle, maintained via several cataplerotic and anaplerotic reactions, is primarily used for biosynthetic processes and not for OXPHOS [249]. Glycolysis and glutaminolysis appear to be further increased in SARS-CoV-2-infected astrocytes [241]. This astrocyte-specific cell metabolism resembles that of other cells (discussed before) that support SARS-CoV-2 replication. Neurons, on the other hand, possess a highly oxidative metabolism, fully depending on OXPHOS. Astrocytes support the delivery of essential substrates to neurons, especially lactate and Glu. In neurons, lactate serves as an important energy source for OXPHOS, and Gln is converted into the neurotransmitters Glu and γ-aminobutyric acid (GABA) [249]. In contrast to astrocytes, the metabolism of neurons under in vivo conditions is thereby focused on energy and neurotransmitter production. This type of metabolism is obviously unsuitable for the replication of SARS-CoV-2. However, the situation probably changes when the neurons derive in vitro from pluripotent stem cells. The culture media used, which contain a rich nutrient supply and antibiotics (e.g., streptomycin), can affect metabolic activity [250,251,252] and could lead to an altered metabolism of the in vitro generated neurons that is better suited to SARS-CoV-2 replication. Together, there is still only limited data on whether and how brain cells adapt their metabolism due to SARS-CoV-2 infection in vivo.

### 4.6. SARS-CoV-2 Infection of the Kidneys

The kidney is a metabolically highly active organ with many specialized EpCs. The nephron is the minute structural and functional unit of the kidney, and it is responsible for the filtration, reabsorption, and secretion of substances from the blood. Essential cells of the nephron are proximal tubular EpCs, distal tubular cells, and podocytes [253]. Major targets for SARS-CoV-2 in the kidney appear to be proximal tubular cells (PTCs) and podocytes [254,255]. The expression of the ACE2 receptor for SARS-CoV-2 was observed in proximal tubule cells and podocytes of the kidney [256,257,258]. Using human pluripotent stem cell-derived podocytes, Kalejaiye and colleagues [257] identified not only ACE2 but also BSG/CD147 as mediators of SARS-CoV-2’s entry into these podocytes, and they showed the direct viral infection of these cells in vitro. SARS-CoV-2 infections in vivo were also reported in proximal tubular EpCs and podocytes, but hardly in distal tubules [231,254,255]. To carry out their specific functions, PTCs are dependent on FAO for energy production, and the acetyl-CoA generated is used for ATP production via OXPHOS. As alternative energy sources, PTCs can also use other substrates like Gln, Pyr, or ketone bodies. Glucose is then mainly generated via gluconeogenesis (GN) [259,260]; glycolysis is almost absent in this state. However, PTCs of injured kidneys can switch from this kind of highly oxidative metabolism (type A) to an alternative metabolism (type B) that uses as the major substrate glucose and glycolysis as the main pathway for energy production [260,261]. This metabolic switch could also be critical for the SARS-CoV-2 infection of PTCs. Based on our previous discussion, type B but not type A metabolism appears to favor SARS-CoV-2 replication. A switch to type B metabolism in PTCs seems to occur in patients suffering from an acute kidney injury (AKI) or a chronic kidney injury (CKI) [68,255]. Indeed, these patients showed a higher rate of kidney infections with SARS-CoV-2 and subsequent severe kidney damages [255,262,263]. An alternative possibility for a switch from type A to type B metabolism could result from the ACE2-independent interaction of the SARS-CoV-2 S protein with TLR4, which leads to cellular activation [264]. Renal tubular cells carry TLR4 on the surface [265] and the interaction with the S protein could lead to this metabolic change [266]. Renal podocytes have also been described as target cells for SARS-CoV-2 [267]. The viral replication has been clearly demonstrated in podocytes generated in vitro from human pluripotent stem cells, as mentioned above [257,268]. The in vivo infection and replication of SARS-CoV-2 in podocytes is less clear [231,267,268]. Podocytes have a high energy demand for carrying out their specific functions. Their metabolism can be driven by OXPHOS or glycolysis, depending on the environmental and nutritional conditions [269]. It is, therefore, possible that SARS-CoV-2 encounters podocytes that have the metabolic requirements for SARS-CoV-2 replication. The podocytes produced in vitro can probably serve as target cells for SARS-CoV-2 since the culture conditions for the generation of these podocytes lead to a cell metabolism that is suitable for the replication of SARS-CoV-2 (see above).

### 4.7. SARS-CoV-2 Infection of the Liver

The liver contains multiple cell types, including hepatocytes, cholangiocytes (EpCs specialized for the modification and transport of bile), stellate cells, specialized MPs called Kupffer cells, and EnCs. The predominant cell type are hepatocytes, which are responsible for carrying out most of the liver’s metabolic functions, including GN, glycogen production, glucose uptake and glycolysis, and lipogenesis [270]. This means that hepatocytes can produce glucose and store it or take up glucose and degrade it, depending on the actual requirement. Thus, the basic metabolism within the hepatocyte population of the liver may be rather heterogeneous [271]. Massive metabolic changes also occur in the hepatocytes of individuals suffering from liver diseases, e.g., fatty liver or liver infections [270]. The expression of ACE2 has been shown in cholangiocytes and, to a lesser extent, in hepatocytes [272,273,274]. The infection of cholangiocytes by SARS-CoV-2 could be detected using human liver ductal organoids [275], but the metabolic state of the infected cells remains unknown. Hepatocytes are mainly discussed as possible important liver target cells for SARS-CoV-2 [276,277,278]. Using autopsy samples from patients who died from SARS-CoV-2 infection, Wanner et al. [277] detected viral RNA not only in the respiratory tract and (at lower levels) in the kidneys, heart, and brain, but also in the liver. The replication-competent virus could also be recovered from liver tissue, suggesting that hepatocytes may support SARS-CoV-2 replication. This is not unexpected, given the above-mentioned metabolic heterogeneity of hepatocytes in the liver. In addition, massive changes to a Warburg-like metabolism, can occur in the hepatocytes of diseased people who suffer, e.g., from a fatty liver or viral infections, especially from HCV [270]. The latter hepatocytes could serve as target cells for SARS-CoV-2 in patients with severe COVID-19 [277,278]. In addition, Barreto et al. [278] reported that the infection of hepatocytes in COVID-19 patients correlates with hyperglycemia, and primary hepatocytes infected with SARS-CoV-2 in vitro showed increased glucose production associated with an increased activity of phosphoenolpyruvate carboxykinase (PEPCK), a central enzyme in GN. The metabolic background of these in vitro cultured hepatocytes remains unknown, but one can assume that these hepatocytes are also metabolically heterogeneous in terms of their glucose metabolism [279]: some cells (type 1) may carry out a more gluconeogenic metabolism, and others (type 2) a more glycolytic metabolism. Type 2 cells may thus be targets for SARS-CoV-2. Metabolic stress conditions, including SARS-CoV-2 infection, stimulate the formation and secretion of Golgi protein 73 (GP73), a secreted hormonal factor that increases hepatic glucose production by inducing the expression of gluconeogenic genes [280]. GP73, expressed in the SARS-CoV-2 infected type 2 cells, could induce glucose production in the type 1 cell population of in vitro cultured primary hepatocytes. Similar events could also occur in the hepatocyte population of SARS-CoV-2-infected patients, leading to hyperglycemia. However, it must be mentioned that SARS-CoV-2 infection in patients can also result in a pancreatic impairment associated with reduced insulin production (infection-induced diabetes).

## 5. Metabolic Aspects of Persistent SARS-CoV-2 Infections

Most patients infected with SARS-CoV-2 are virus-free within 2-4 weeks when tested for viral antigens or RNA [281]. However, in a minority of COVID-19 patients, viral RNA remains detectable in the respiratory tract and other tissues and persists there for many weeks, with or without the shedding of infectious virus particles [232,282,283,284,285]. Persistent infections, associated with a continuous spread of the virus, have been observed, particularly in immunocompromised patients, which indicates that an intact immune system may inhibit the emergence of persistence [286,287]. The occurrence of persistent infections may have severe consequences: it has been argued that such infections may contribute to the post-acute sequelae of SARS-CoV-2 infection (PASC), also known as long COVID-19 syndrome [288,289,290,291]. On the other hand, even in immune-competent patients, persistent viral RNA can be frequently observed without the shedding of infectious virus [292,293]. We propose that the difference in the occurrence of SARS-CoV-2 persistence in immunocompetent and immunocompromised individuals is not only due to the different immune status of the affected individuals but may also depend on the respective metabolic conditions, as discussed below. To study persistent SARS-CoV-2 infections at the cellular level, Gamage et al. [163] used in vitro differentiated nasal EpCs (NECs) and bronchial EpCs (BECs), each infected with SARS-CoV-2. In both model systems, SARS-CoV-2 could be detected in acutely (3 dpi) and persistently (28 dpi) infected cell populations. Antiviral cellular responses, including the induced expression of type I and type III IFNs and, subsequently, of ISGs, can equally occur in both cell types; i.e., differences in cell-intrinsic antiviral responses is not the reason for the occurrence of persistence. Infectious particles are produced in both cell populations, which means that the viral proteins ORF3b and ORF6 are also produced, interrupting IFN-signaling pathways [61,294]. However, like what has been discussed above, the two differentiated NEC and BEC populations may exhibit intrinsic metabolic heterogeneity; i.e., only some cells may carry out a glycolysis-based metabolism suitable for SARS-CoV-2 replication (type 1 cells). Most cells (type 2 cells) may have an OXPHOS-driven metabolism, which is typical of differentiated quiescent EpCs but is unfavorable for efficient SARS-CoV-2 replication. Type 1 cells can take up SARS-CoV-2 via the ACE2 receptor and may allow for the efficient, acute viral replication and shedding of infectious particles. Type 2 cells may take up SARS-CoV-2, secreted by type 1 cells, either via the ACE2 receptor or via transmission through cell-to-cell contact with type 1 cells [295]. The latter mode of the endosomal internalization of the SARS-CoV-2 virus depends also on the viral S protein, but it does not necessarily require ACE2 [295]. Instead, S protein-mediated cell-to-cell contact can occur via a receptor (e.g., TLR4), which can also induce a change in the metabolism in the type 2 cell (e.g., via the activation of NF-kB and/or mTOR). This metabolic change can lead to partial or complete virus replication corresponding to persistent infection. In some of the infected NECs and BECs, the virus or its genomic RNA will remain enclosed in the endosome. Together, the spectrum of events observed upon the infection of differentiated EpCs by SARS-CoV-2, leading to short- and long-term (”persistently”) infected cells carrying high or lower viral loads or viral RNA only, can be explained by the probable expected metabolic changes in these cell populations, i.e., in the absence of a functional immune system. Similar processes may occur in the SARS-CoV-2-infected tissues and organs of immuno-compromised patients, and this will lead to the persistence of the virus or viral components in some cells, as observed in such patients [286,287]. 

The conditions for the emergence of SARS-CoV-2’s persistence in immunocompetent individuals are, of course, different. The metabolic conditions for infection and spread of SARS-CoV-2 in a tissue are initially similar in immunocompetent and immune-compromised individuals; i.e., some differentiated tissue and organ cells can take up and replicate the virus. However, when infectious viruses are released from these cells, the free viruses are neutralized by the presence of specific antibodies before they reach neighboring cells, where they eventually persist. But also, in this case, SARS-CoV-2 can be transmitted through cell-to-cell contact into neighboring cells, where the virus may multiply or persist—depending on the metabolic state of the recipient cell. However, these cells carrying the intact virus may be recognized and killed by cytotoxic T cells (CD8^+^ T cells). In contrast, in cells that only carry viral genomes, T cell antigens will not be recognized by the T cells. In this way, virus RNA can persist even in immunocompetent individuals [232,296,297].

## 6. Discussion and Future Directions

The efficient formation of infectious SARS-CoV-2 particles requires large amounts of nucleotides, amino acids, and fatty acids/lipids, which must be provided by the host cell to produce the virus-specific RNAs, proteins, and lipids. This means that the virus, once internalized (in the case of SARS-CoV-2, mainly through the interaction of its spike protein S with the ACE2 receptor of the target cell), must either encounter a pre-existing cellular metabolism that is already appropriate for the viral replication, or the virus must reprogram the infected cell’s metabolism to meet the specific needs for efficient virus replication. The cellular metabolism allowing SARS-CoV-2 replication and multiplication has been studied in vitro using several established cell lines, cultured primary cells, and different organoids, as well as in vivo using tissue cells of different organs from COVID-19 patients. Most of the used cell lines (often cancer cell-derived) perform a ”Warburg-like“ metabolism. In addition, the culture media used for the growth of these host cells contain nutrients for virus replication and usually antibiotics (e.g., streptomycin), which can disrupt mitochondrial functions [298] and thus promote glycolysis. These metabolic conditions appear to be favorable for SARS-CoV-2 replication, and not much metabolic reprogramming by the virus seems to be required to make the metabolism of these target cells suitable for efficient SARS-CoV-2 multiplication. The metabolic situation of organoids, which are often used to study SARS-CoV-2 replication in vitro, is more difficult to assess. Organoids, generally produced from multipotent stem cells, resemble natural organs morphologically and functionally more than cultured (primary) cells. However, their correspondence with the metabolic realities of organs is doubtful. The cultivation of the stem cells, as well as their differentiation into organ-specific cells, generally takes place in rich culture media and in the presence of antibiotics that may affect the cell metabolism (e.g., using aminoglycosides). These conditions could significantly affect the metabolism in the organoid cells and, thus, their susceptibility to infection by SARS-CoV-2.

Under real in vivo conditions, i.e., in SARS-CoV-2-infected patients, the situation is extraordinarily complex as far as the metabolism of possible (ACE2^+^) target cells is concerned. From the available data, it is obvious that not only the level of ACE2, together with TMPRSS2, is decisive for the rapid invasion and efficient replication of the virus within target cells of the different organs that can be infected by SARS-CoV-2, but so is an appropriate metabolism of the infected cells. The susceptibility of an individual to the virus will, therefore, depend (apart from immune status, which is not discussed here) on the overall physiological and nutritional status of the infected individual, which can also lead to a different metabolism of the potential target cells for SARS-CoV-2, such as in diabetics, obese individuals, or patients with liver diseases. When SARS-CoV-2 is internalized by differentiated cells of a specific organ through receptor-induced endocytosis, it will meet only a few tissue cells that are likely to (stochastically) carry out a metabolism suitable for the proliferation of SARS-CoV-2. But most differentiated cells of tissues, potentially infected with SARS-CoV-2, lack the sort of metabolism that appears to be optimal for efficient SARS-CoV-2 multiplication. The metabolism of these tissue cells is normally geared towards specialized functions that often require a high level of energy (ATP) that is gained through OXPHOS in the mitochondria, while the anabolic pathways leading to metabolites needed for virus production are less active. This means that, even if viruses are taken up by such differentiated ACE2^+^ cells, they may not be able to proliferate efficiently in these cells without reprogramming their metabolism to a state suitable for viral multiplication. As outlined in this review, there are obviously several ways in which SARS-CoV-2 can achieve this goal:(i)The few primarily infected tissue cells (stochastically performing a metabolism suitable for virus replication) may secrete cytokines (e.g., IL-1 and TNF-α) that can convert the metabolism of neighboring resting cells such that it becomes suitable for SARS-CoV-2 replication.(ii)The S protein of the released SARS-CoV-2 viruses can interact with receptors such as GRP78, MR, DC-sign, and especially TLR4, which trigger signaling pathways (e.g., PI3K/Akt/mTOR) and regulatory factors (e.g., HIF-1α) that activate glucose uptake, glycolysis, and anabolic pathways but suppress Pyr’s entry into the mitochondria and, thus, OXPHOS.(iii)The virus can also actively contribute to the establishment of a pro-viral host-cell metabolism through the production of specific NSPs and ACPs that appear to modify and/or stabilize cell metabolism, which is already predisposed to SARS-CoV-2 replication. Examples are the blockade of the activity of p53 via NSP5, the activation of HIF-1α via ORF3a, the activation of NF-kB via ORF7a and other viral proteins—and in particular by hijacking the host-cell protein machinery by blocking the access of the host-cell mRNAs, but not that of the viral mRNAs, to the ribosome with the help of NSP1. The untranslated host-cell mRNAs are degraded and can serve as a nucleotide pool for SARS-CoV-2 transcription and replication.(iv)Other virus-specific proteins might inhibit the entry of Pyr into the mitochondria or the conversion of Pyr into acetyl-CoA through activation of PDHK. The latter (still hypothetical) reactions could inhibit OXPHOS and favor (aerobic) glycolysis in infected differentiated cells, making them suitable for SARS-CoV-2 replication. In addition, it should be mentioned that antiviral cell activity can also be blocked via SARS-CoV-2-encoded proteins, especially NSP1 and ORF6, the two most potent SARS-CoV-2 inhibitors of type I (and type III) IFNs, and the subsequent IFN-induced expression of ISGs, which can block the viral replication cycle [61,299].

Overall, the host-cell metabolism, which is likely to be optimal for the efficient multiplication of SARS-CoV-2, appears to be based on the function of catabolic and anabolic pathways (see also Figure 1). Recent progress in multi-omics technologies, such as in the analysis of interactomes, opens new avenues for a better understanding of the interplay between the virus and its host [300,301]. However, it must be recognized that the precise metabolism of host cells suitable for the replication of SARS-CoV-2 and other viruses is difficult to determine using currently available methods. The conditions of in vitro cell cultures, even when primary tissue cells or organoids are used, hardly reflect the conditions of differentiated cells in an infected organ. Single-cell analyses, which would be necessary to precisely determine the metabolic processes of an infected cell, are limited and currently partly possible using single-cell functional genomics, including the organelle profiling of virus-host interactions [302], mainly based on single-cell RNA-seq. By comparing the level of mRNAs, which code for enzymes involved in metabolism from infected and uninfected cells, plus/minus statements can be made. However, the actual course and dynamics of the cell metabolism during a SARS-CoV-2 infection can hardly be deduced from such data due to (i) the different stability of the mRNAs, (ii) possible posttranscriptional modifications of mRNAs, (iii) the different translational efficiency of the mRNAs (see, e.g., the function of NSP1), and (iv) post-translational modifications of metabolic enzymes. Metabolomics [303,304,305] and, in particular, the use of ^13^C-isotopologue analysis offer considerable advantages here [306,307,308]. However, the current sensitivity of metabolome analysis and the ^13^C-isotopologue method is a challenging limit to these techniques on a single-cell basis. In conclusion, further progress in analytics is needed to precisely determine the cell metabolism in vivo on a single-cell basis, which would be necessary to understand the onset and progression of not only viral but also other intracellular microbial infections. In our opinion, this understanding is also an indispensable prerequisite to safely implementing therapeutic measures against SARS-CoV-2 infections directed against host metabolic targets, as suggested in previous publications [19,70].

## Figures and Tables

**Figure 1 ijms-25-09977-f001:**
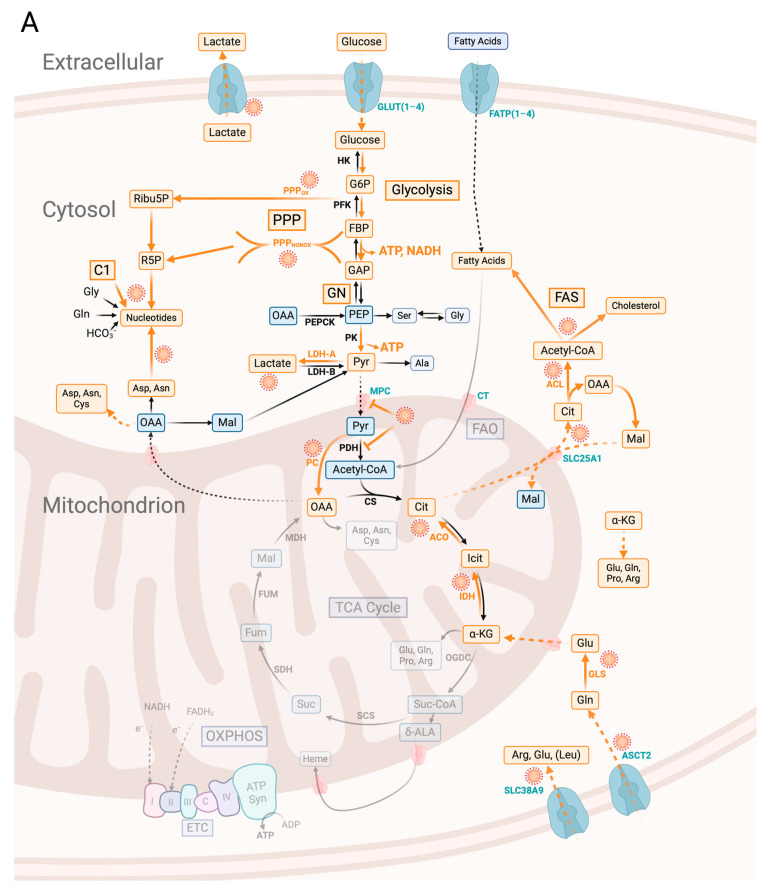
(Created with BioRender.com): (**A**) Metabolic program of SARS-CoV-2-infected cells suitable for viral replication (as derived from the discussed studies on SARS-CoV-2 replication in established cell lines and tissue cells). The major metabolic reactions supporting this program are marked by the virus symbol and broad orange arrows, and they include the following: (a) increased glucose uptake via the induction of glucose transporters, especially GLUT1 and GLUT3; (b) enhanced (aerobic) glycolysis due to increased PKM2, combined with increased lactate dehydrogenase A (LDH-A) activity, to regenerate NAD for the maintenance of glucose oxidation via glycolysis, which is necessary for the increased production of ATP, glucose-6-phosphate (G6P) (required for the initiation of PPP), and 3-phosphoglycerate (synthesis of serine/glycine, erythrose-4-phosphate); (c) activation of PPP with both arms to generate ribose-5-phosphate (R5P) and NADPH; (d) reduced OXPHOS via the inhibition of the pyruvate dehydrogenase (PDH)-mediated mitochondrial acetyl-CoA formation, which could be achieved either through the inhibition of the mitochondrial pyruvate carrier complex (MPC), which would block the entry of pyruvate (Pyr) into the mitochondria, or through the activation of pyruvate dehydrogenase kinase (PDHK), which would block the conversion of Pyr into acetyl-CoA; (e) specific metabolites normally produced in the TCA cycle and indispensable for SARS-CoV-2 replication, with α-KG, OAA, acetyl-CoA, citrate (Cit), and Suc-CoA especially able to be delivered via active TCA-cycle enzymes or various anaplerotic reactions, as α-KG can be provided via glutaminolysis, OAA via ATP-dependent PC and ATP-dependent Cit lyase (ACL), acetyl-CoA via fatty acid oxidation (FAO), Cit either via TCA-cycle enzymes from acetyl-CoA and OAA or reverse TCA-cycle reactions through the reduced carboxylation of α-KG and Suc-CoA via the TCA cycle enzyme α-KG dehydrogenase; (f) in the case of a glucose shortage, FAO is an alternative route for ATP generation. (**B**) Viral reprogramming of cellular regulators that modify metabolic fluxes in favor of SARS-CoV-2 replication. Metabolic reactions that are positively or negatively regulated through viral mechanisms are shown in orange boxes. Viral factors partaking in host-cell control are shown in orange, oval boxes indicated by a virus particle. For details and a further legend, see also Appendix A.

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
