# Peer review of "Interactions of SARS-CoV-2 with Human Target Cells—A Metabolic View"

_ijms, 2024, doi:10.3390/ijms25189977_

Round 1

Reviewer 1 Report

Comments and Suggestions for Authors

Estimated Editors,

I've been invited to peer-review this narrative review from Eisenreich et al. entitled "Interactions of SARS-CoV-2 with Human Target Cells – A Metabolic View".

In this study, well written, well organized, but also very well documented through an up-to-date and critically discussed bibliography, Authors report what we know (at the moment) about SARS-CoV-2 and its interaction with various tissues from the Human body, providing a summary that is both wholesome but also effective in providing a reliable summary.

The text is also enriched by two pictures that provide a very accurate outline of the potential interactions of SARS-CoV-2.

Even though the text is very long, I think that it does not require or deserve any request for shortenings or summarizing.

From my point of view, the paper could be accepted as it is.

Author Response

We thank the reviewer for his/her encouraging comments.

Reviewer 2 Report

Comments and Suggestions for Authors

Interactions of SARS-CoV-2 with Human Target Cells – A Metabolic View

In this review, authors have attempted to provide the latest updates on the metabolic view of the interaction of SARS-CoV-2 with human target cells. Overall, this review summarizes what is known about the metabolic regulation of human host cells in the case of SARS-CoV-2 infection. The current review is well-formulated and nicely written. Following are the specific comments to further strengthen the manuscript,

1.        It looks like it is missing the name of the smallest genome virus 7.5 kb. Consider providing this information in the introduction on line number 34.

2.        Line number 55: is it a complex network? Consider changing net to network.

3.        Figure quality may be improved for better visualization.

4.        If information is available, it will be interesting to add some information on the metabolic reprogramming of microglial cells upon SARS-CoV-2 infection. Or can it be extrapolated from the info on metabolic reprogramming of the monocyte/macrophages? This information will help better understand the long COVID perspective.

5.        Is there any study to compare the metabolic reprogramming of host cells after vaccination? It will be interesting to provide a comparative view of metabolic reprogramming between vaccinated and non-vaccinated scenarios.

Author Response

  1. It looks like it is missing the name of the smallest genome virus 7.5 kb. Consider providing this information in the introduction on line number 34.

We have changed this sentence (line 33) into: "While the genome size of many +ssRNA viruses including the Polio Virus and Hepatitis E Virus is only about 7.5 kb, that of the three ß-Coronaviruses mentioned above (the largest among the +ssRNA viruses) is approximately 30 kb."

  1. Line number 55: is it a complex network? Consider changing net to network.

We have changed net into “network”.

  1. Figure quality may be improved for better visualization.

If necessary, we can submit png files at the highest possible resolution during the processing of the article.

  1. If information is available, it will be interesting to add some information on the metabolic reprogramming of microglial cells upon SARS-CoV-2 infection. Or can it be extrapolated from the info on metabolic reprogramming of the monocyte/macrophages? This information will help better understand the long COVID perspective.

There is rather limited information about metabolic reprogramming of brain cells. In the revised chapter 4.5., we have now reviewed in more detail the current knowledge about the infectivity and responses of brain cells, with a special focus on microglial cells and their metabolic phenotype. Here, we have now included two more recent publications (Andrews et al. 2022 and Jeong et al. 2022) dealing with this topic.

          “A relatively large proportion of COVID-19 patients suffer from neurological symptoms, suggesting that the central nervous system (CNS) can be invaded by SARS-CoV-2. The most likely access to the CNS is the viral spreading through the olfactory mucosa thus causing brain infections [227,228]. Indeed, autopsies of patients who died of COVID-19 showed sub-genomic SARS-CoV-2 RNA in the CNS, which is a sign for active viral replication in brain cells [229,230]. There are two main cell types present in the brain that are also the likely candidates for SARS-CoV-2 infections: neurons and non-neuronal glial cells (astrocytes, microglia, and oligodendrocytes). Astrocytes and microglia are the most abundant cells in the brain. Astrocytes are mainly involved in supporting neuronal functions, while microglial cells are the dominant resident immune cells acting as the immune system of the brain. Both, neurons and glial cells carry out distinct, but partially interacting metabolic processes that are essential for the proper brain function [231].

Studies regarding SARS-CoV-2 infection of brain cells have been performed with various neural models, including post-mortem specimen of SARS-CoV-2 patients with neurological symptoms, human pluripotent stem cell-induced neural cell models, such as pure neurons, astrocytes, and microglia, as well as 3D organoids. The results of most of the earlier studies show that it is not clear or sometimes even contradictory [232,233] which cells of the brain are susceptible to SARS-CoV-2 [234]. Expression of ACE2 has been detected in cultured astrocytes and microglial cells but not in neurons, suggesting that astrocytes might be target cells for SARS-CoV-2 [235-237]. However, pluripotent stem cell-derived neurons also express ACE2 [228], and studies with 3D human brain organoids derived from pluripotent stem cells provided evidence that SARS-CoV-2 infects neurons, but not astrocytes [238,239]. The data on in vivo expression of ACE2 on glial cells and neurons are also still rather ambiguous [234,236,237]. In a recent study using in vivo brain samples from individuals with neurological symptoms (who died of SARS-CoV-2 infections) and in vitro cultured human neural stem cell-derived astrocytes infected with SARS-CoV-2, the authors showed that astrocytes are the main site of infection (and probably replication) for SARS-CoV-2 in the brain [240]. Interestingly, they also showed that the astrocytes do not express ACE2, but efficient uptake of the virus occurs by the interaction of the spike protein with neuropilin-1 (NRP1), an alternative receptor for SARS-CoV-2 [241]. Preferential infection of astrocytes by SARS-CoV-2 was also reported by Andrews et al. in 2022 [242] using cortical organoids derived from human pluripotent stem cells (hPSCs) as well as from primary human cortical tissues. However, neurons and microglia did not seem to be directly infected by SARS-CoV-2. In contrast, human microglial clone 3 cells (HMC3) were infected by SARS-CoV-2, as shown by Jeong et al. in 2022 [243]. This virus-infected cell line showed a M1 phenotype which may support the replication of SARS-CoV-2 as described above for the replication of the virus in M1 macrophages. However, the authors also showed that, in K18-hACE2 transgenic mice, the microglia were also infected by the virus upon intranasal inoculation of SARS-CoV-2.

These partially ambiguous in vitro and in vivo results concerning infection of brain cells by SARS-CoV-2 are not too surprising if one considers the complex metabolic interactions between the brain cells and in particular between astrocytes and neurons that occur under in vivo conditions [231,244,245]. Astrocytes are mainly glycolytic cells which can even survive inhibition of mitochondrial respiration by performing enhanced aerobic glycolysis [245]. Glucose is the preferred carbon and energy substrate, which is channeled into glycolysis and PPP, while entry of Pyr into the TCA cycle is limited due to reduced PDH activity caused through phosphorylation by enhanced PDHK activity [246,247]. Excess Pyr is converted to lactate which is secreted and serves as a substrate for neurons (see below). The TCA cycle, maintained by several cataplerotic and anaplerotic reactions, is primarily used for biosynthetic processes, and not for OXPHOS [248]. Glycolysis and glutaminolysis appear to be further increased in SARS-CoV-2 infected astrocytes [240]. This astrocyte-specific cell metabolism resembles that of other cells (discussed before) that support SARS-CoV-2 replication. Neurons, on the other hand, possess a highly oxidative metabolism, fully depending on OXPHOS. Astrocytes support the delivery of essential substrates to neurons, especially lactate and Glu. In neurons, lactate serves as an important energy source for OXPHOS, and Gln is converted to the neurotransmitters Glu and g-aminobutyric acid (GABA) [248]. In contrast to astrocytes, the metabolism of neurons under in vivo conditions is thereby focused on energy and neurotransmitter production. This type of metabolism is obviously unsuitable for the replication of SARS-CoV-2. However, the situation probably changes when the neurons derive in vitro from pluripotent stem cells. The culture media used, which contain a rich nutrient supply and antibiotics (e.g., streptomycin), can affect metabolic activity [249-251] and could lead to an altered metabolism of the in vitro generated neurons that is better suited for SARS-CoV-2 replication. Together, there is still only limited data if and how brain cells adapt their metabolism due to SARS-CoV-2 infection in vivo.”

  1. Is there any study to compare the metabolic reprogramming of host cells after vaccination? It will be interesting to provide a comparative view of metabolic reprogramming between vaccinated and non-vaccinated scenarios.

To the best of our knowledge, there is no conclusive study about the metabolic reprogramming of infected non-immune host cells upon vaccination.

Reviewer 3 Report

Comments and Suggestions for Authors

In this paper, the authors review published literature to explore how SARS-CoV-2 influences cellular metabolism, with a specific focus on glucose metabolism. They first outline the key steps of viral infection and discuss how viral proteins impact each step and associated cellular metabolic pathways. The authors then integrate current knowledge on how the virus affects metabolism in cell lines or in vitro models and extend their discussion to how it impacts organ-level metabolism. The paper concludes by examining the potential relationship between chronic infection and long-term metabolic changes.

Given that diabetes represents a classic case of metabolic dysregulation, it is somewhat disappointing that the authors did not address the relationship between this condition and viral infection. Additionally, some ongoing debates in the field, such as the reasons behind increased viral susceptibility in diabetic patients and the differential expression of ACE2 in various types of nasal epithelial cells (ciliated epithelial cells and goblet cells), are not fully explored from both perspectives.

Nevertheless, the authors' synthesis of the literature and their reasonable hypotheses provide significant contributions to this field. I also find their perspective on how carbohydrate metabolism influences viral tropism to be both important and intriguing. For researchers looking to delve into viral metabolism, this paper serves as an essential entry point. Below are some suggestions for further improvement.

1.      The authors have compiled the effects of viral infection on metabolism across different cell types and tissues. I recommend creating separate tables under the sections for cell lines and tissues. These tables could include information such as viral susceptibility, primary metabolic pathways of the cells, and specific metabolic pathways affected by the virus. Such comparative tables would enable readers to quickly assess whether the inherent metabolic state of the cell truly influences viral infection and how the virus, in turn, impacts cellular metabolism.

2.      In line 486 and 899, the term 'Warburg-like' metabolism is commonly used in cancer metabolism research to describe a metabolic shift where cells rely more on glycolysis than oxidative phosphorylation for energy production, even in the presence of oxygen. However, since this term may not be as familiar to scholars in the field of virology, it’s important to briefly explain that in this context, it refers to similar metabolic changes observed in virus-infected cells, where cells shift to glycolysis to meet the increased energy and biosynthetic demands during viral replication.

3.      In line 119, why do the double membrane vesicles formed during viral replication affect amino acid or lipid metabolism? Is it due to the extensive consumption of membrane components? The connection between these processes hasn’t been clearly described.

4.      In line 120 and 121, what is the definition of disrupted lipid metabolism? What are the indicators? This has not been clearly described.

5.      In line 845, changing 'respiratory EpCs (AECs)' to 'bronchial EpCs (BECs)' would be more accurate. The original article also uses this terminology. Otherwise, when I see 'respiratory EpCs,' I assume you are referring to alveolar epithelial cells.

Author Response

Given that diabetes represents a classic case of metabolic dysregulation, it is somewhat disappointing that the authors did not address the relationship between this condition and viral infection. Additionally, some ongoing debates in the field, such as the reasons behind increased viral susceptibility in diabetic patients and the differential expression of ACE2 in various types of nasal epithelial cells (ciliated epithelial cells and goblet cells), are not fully explored from both perspectives.

Answer: In line 317 ff, we have now briefly mentioned the relationship of metabolic deregulation in diabetes and viral infection:

“This could be an important finding as it possibly explains why high cellular glucose levels as evident in diabetes patients lead to increased SARS-CoV-2 infections (for a recent review, see [91]). However, increased rates of SARS-CoV-2 infections in diabetes patients may also result from enhanced uptake of the virus due to low serum levels of 1,5-anhydro-D-glucitol [92]."

In chapter 4.1., we have discussed the differential expression of ACE2 in the various types of nasal epithelial cells.

Reviewer's suggestions:

  1. The authors have compiled the effects of viral infection on metabolism across different cell types and tissues. I recommend creating separate tables under the sections for cell lines and tissues. These tables could include information such as viral susceptibility, primary metabolic pathways of the cells, and specific metabolic pathways affected by the virus. Such comparative tables would enable readers to quickly assess whether the inherent metabolic state of the cell truly influences viral infection and how the virus, in turn, impacts cellular metabolism.

Answer: We did not follow the reviewer´s recommendation to create the proposed tables, since in our opinion there is not sufficient clear-cut information to provide a conclusive view in a simplified table format concerning viral susceptibility, primary metabolic pathways of the cells, and specific metabolic pathways affected by the virus. We think that we have pointed out in detail in the text what is known about these complex aspects.

  1. In line 486 and 899, the term 'Warburg-like' metabolism is commonly used in cancer metabolism research to describe a metabolic shift where cells rely more on glycolysis than oxidative phosphorylation for energy production, even in the presence of oxygen. However, since this term may not be as familiar to scholars in the field of virology, it’s important to briefly explain that in this context, it refers to similar metabolic changes observed in virus-infected cells, where cells shift to glycolysis to meet the increased energy and biosynthetic demands during viral replication.

Answer: In lines 448 ff, we have now introduced the term “Warburg-like” metabolism: “This type of metabolism is also named “Warburg-like” metabolism, well known in cancer cells to describe a metabolic shift where cells rely more on glycolysis than on OXPHOS for fast ATP generation, even in the presence of oxygen.”

  1. In line 119, why do the double membrane vesicles formed during viral replication affect amino acid or lipid metabolism? Is it due to the extensive consumption of membrane components? The connection between these processes hasn’t been clearly described.

 Answer: See answer 4.

  1. In line 120 and 121, what is the definition of disrupted lipid metabolism? What are the indicators? This has not been clearly described.

Answer: In lines 122 ff, we have better described this connection:

“This will affect the host cell´s fatty acid/lipid metabolism due to the increased demand for fatty acids during viral membrane synthesis. However, in particular infections caused by +ssRNA viruses could also generally trigger lipid rearrangements in their hosts [49,50].”

  1. In line 845, changing 'respiratory EpCs (AECs)' to 'bronchial EpCs (BECs)' would be more accurate. The original article also uses this terminology. Otherwise, when I see 'respiratory EpCs,' I assume you are referring to alveolar epithelial cells.

Answer: We have changed AECs into BECs and added BECs into the abbreviation list.